# Evaluation of multiple extraction and inactivation methods for the rapid identification of filamentous fungi using matrix-assisted laser desorption/ionization time-of-flight mass spectrometry

Halimat Olaniyan,[1] Chris Kendra,[1] Alexandria Dugan,[2] Shelby Kruer,[2] Alisha Zuber,[2] Jessica Bywaters,[2] Amber Ryan,[2] Kenneth Gavina[1,2]

**ABSTRACT** Matrix-assisted laser desorption/ionization-time of flight mass spectrometry (MALDI-TOF MS) offers high sensitivity for the rapid identification of microbial isolates. While predominantly used in bacterial and yeast identification within the clinical laboratory, recent advancements have prompted interest in expanding MALDI-TOF MS applications for the rapid identification of filamentous fungi, particularly clinically relevant molds. However, the implementation of mold identification protocols introduces biosafety concerns due to the potential for spore aerosolization during sample handling and inoculation, posing occupational hazards to laboratory personnel. To evaluate the feasibility and safety of MALDI-TOF MS-based mold identification, we conducted a prospective study comparing an extraction-free approach to two different rapid extraction methods across routine media and specialized fungal plates. Each method was assessed for spectral quality, reproducibility, and identification accuracy using a validated reference library. Importantly, we performed a biosafety validation study to assess the efficacy of the extraction-free approach in inactivating mold spores and hyphal elements. The extraction-free approach generally outperformed the rapid extraction methods, demonstrating more accurate fungal identifications (90.5% vs. 61.9% and 66.7%) and a higher average spectra score; accuracy and spectra scores were further enhanced using specialized fungal media (92.9% vs. 81.0% and 52.4%). Our findings support the use of this extraction-free approach as both a safe and effective method to rapidly identify filamentous fungi using MALDI-TOF MS within the clinical laboratory.

**IMPORTANCE** This study overviews the implementation of identifying clinically relevant filamentous fungi using matrix-assisted laser desorption/ionization time-of-flight mass spectrometry (MALDI-TOF MS) in a clinical laboratory. The methodology applied in this study produced both highly sensitive and specific results with significantly reduced turnaround time compared to traditional standard of care culture-based workup. Faster time to detection has the potential to impact the healthcare treatment of millions of patients each year without requiring significant changes to clinical workflow. This study used multiple approaches to improve the clinical utility and performance of MALDI-TOF MS for fungal identification. Importantly, it also highlights the effectiveness of a developed inactivation method to ensure laboratory personnel safety. This method and workflows are of interest to support clinical microbiology diagnostics and to help aid in infection identification to enhance timely treatment.

**KEYWORDS** mycology, MALDI-TOF MS, clinical microbiology, fungal diagnostics

**Peer Reviewer** Anna Frances Lau, National Institutes of Health Clinical Center, Bethesda, Maryland, USA

Address correspondence to Kenneth Gavina, gavinak@iu.edu.

The authors declare no conflict of interest.

Filamentous fungi are ubiquitous environmental organisms, representing a significant proportion of clinically relevant fungal pathogens in immunocompromised hosts. Global epidemiological data estimates an annual incidence of approximately 6.5 million invasive fungal infections, contributing to approximately 3.8 million deaths worldwide (1). In the United States alone, the direct healthcare expenditure related to fungal disease hospitalizations totals 4.6 billion USD (1). Conventional fungal diagnostics rely on phenotypic characterization, including macroscopic colony morphology and microscopic features, which are inherently slow and labor intensive. Traditional culture-based methods are also subject to inter-observer variability, especially in instances of morphologic overlap among closely related taxa (2). While culture-based identification remains the diagnostic gold standard for molds, its reliability is contingent upon technical expertise, and results may be variable by observing technologist (3).

Recent advancements have highlighted the potential of matrix-assisted laser desorption/ionization-time of flight mass spectrometry (MALDI-TOF MS) to facilitate the rapid and accurate identification of filamentous fungi (2, 4, 5). MALDI-TOF MS has already revolutionized clinical laboratory workflow efficiency and species-level resolution in both bacterial and yeast diagnostics (6). Its application to mold identification offers potential for enhancing species-level accuracy, reducing diagnostic latency, and minimizing the need for specialized training of laboratory personnel; however, successful implementation is dependent on updated, robust spectral databases and safe, optimized protein extraction methods (2). Endemic dimorphic fungi, which exhibit thermal dimorphism growing as yeasts at physiological temperatures and as molds at ambient temperatures, are of particular concern due to their pathogenicity and are among the leading causes of laboratory-acquired infections in the United States (7). Standard laboratory practice for handling presumptive dimorphic fungi requires Biosafety Level 3 (BSL-3) containment (8). Traditional direct smear techniques, commonly used for bacterial and yeast identification, are inadequate and pose occupational hazard concerns for lab personnel (9), making the inactivation of mold extraction protocols critically important for laboratory safety. Multiple full and rapid mold extraction protocols have been published (3, 10, 11), but mold inactivation has not been extensively described. In this study, we systematically evaluated different extraction and inactivation protocols for their safety, accuracy, and feasibility in the application of MALDI-TOF MS for mold identification. In addition to the inactivation study, we compared the different inactivation methodologies using both routine media and specialized conidiation fungal plates to assess which workflow produced the highest spectral scores and most accurate identifications.

## MATERIALS AND METHODS

### Reference strains and clinical isolates

This study utilized both clinical fungal specimens and reference strains obtained from the American Type Culture Collection. Clinical specimens were obtained between January 2022 and December 2024 from the Sidney and Lois Eskenazi Hospital and Indiana University Health Pathology Laboratory in Indianapolis, Indiana. A total of 42 unique filamentous fungal species were sub-cultured onto both solid and liquid routine media to serve as growth controls and evaluate media-dependent variability in protein extraction and MALDI-TOF MS identification. Fungal identification of these clinical isolates was determined through a combination of standard-of-care workup and fungal sequencing. Technologists running the tests were all trained on how to optimize biomass collection with standardized harvest from the colony edge, targeting approximately 1 cm on solid media or equivalent mycelial mass from liquid broth.

### Fungal culture media and protein extraction protocols

Fungal isolates were inoculated onto a variety of routine media, including Sabouraud Dextrose Agar (SAB), SAB + Gentamicin (SABG), Potato Dextrose Agar, Brain Heart

Infusion Agar, and SAB broth. Fungal isolates were also plated on ID-Fungal Plates (IDFP; Conidia, FR), a selective conidiation medium that facilitates the harvesting of fungal mycelia.

Three distinct methodologies were performed and evaluated for MALDI-TOF MS analysis, including the Bruker Mycelium Transfer (MyT), a manufacturer-developed, rapid extraction-free procedure, and two full rapid extraction methods from the National Institutes of Health (NIH) and the Centers for Disease Control and Prevention (CDC) (3, 10–12). Briefly, both full extraction protocols utilize an ethanol-acetonitrile-based chemical extraction approach while employing silica beads for mechanical disruption of the fungal cell wall (11). The extracted proteins were then spotted onto a MALDI target plate, overlaid with a matrix, and analyzed via mass spectrometry to generate spectra for database comparison and identification. For the MyT method, a sterile toothpick applicator was pre-treated with 70% formic acid by dipping the tip into an Eppendorf tube (10). Mycelium was then harvested from the encroaching edge of each of the fungal cultures, ensuring that an equivocal sample was collected for the controls, and then spotted onto the MALDI target plate. All procedures were performed with appropriate personal protective equipment (PPE) in a biological safety cabinet (BSC) within a BSL-2 laboratory; additionally, potential high-risk aerosolizing fungal pathogens (e.g., dimorphic molds) were worked up within BSL-3 containment.

## Inactivation study design

To evaluate the inactivation effectiveness of the MyT method, a 4-week study was established using 42 unique fungal specimens. All sample prep and manipulation were performed in a secure area of the lab under a biosafety hood to minimize personnel exposure. Samples were tested in triplicate on solid and liquid media by the same operator. All operators were trained to standardize biomass to approximately 1 cm. For inoculation onto solid media, mycelium was sub-cultured onto a new plate, and 1 µL of formic acid was added as an overlay at the implantation site. For liquid media, mycelium was harvested, and a 1 µL overlay of formic acid was added directly to the applicator and allowed to air-dry before introducing the inoculum to the liquid broth. The process was repeated for all unique fungal specimens, and each specimen was incubated for 28 days, assessed weekly, to observe for any breakthrough growth. Additionally, control solid and liquid media were sub-cultured in parallel for each fungal specimen to ensure fungal viability. In cases where breakthrough growth was observed, the MyT method was repeated by two independent operators, and the cultures were re-inoculated. Replicates were then incubated for an additional 4 weeks, and the best result out of three was recorded.

## Assessment of different protocols by MALDI-TOF MS

A total of six protocol iterations were conducted to determine the optimal workflow for maximizing MALDI-TOF MS identification accuracy while ensuring biosafety: (i) MyT method from routine culture; (ii) NIH extraction method from routine culture; (iii) CDC extraction method from routine culture; (iv) MyT method from IDFP; (v) NIH extraction method from IDFP; and (vi) CDC extraction method from IDFP. MALDI-TOF MS was performed on the Bruker MALDI Biotyper following the manufacturer's instructions for set-up. Mass spectrometry data were generated using automated spectra generation and analyzed using the Bruker MBT Compass software (V4.1.100) with the research-use-only (RUO) HT Filamentous Fungi CA Module (12). Identification scores were compared across different media types, extraction protocols, and inactivation methods to determine the most effective workflow for accurate and safe mold identification via MALDI-TOF MS.

## Statistical analysis

Statistical analyses and graphical presentations were generated using Prism V9.3.1 (GraphPad, San Diego, CA). For comparing the performance of the MALDI-TOF MS

identification with the different extraction methods, we performed a Kruskal-Wallis analysis, assuming non-normal distribution, with a Dunn's multiple comparisons test; significant differences were determined with an alpha value of 0.05 and were graphically represented with a box-and-whisker plot.

## RESULTS

### Inactivation study

A summary of results from the inactivation arm of the study can be found in Table 1. All control isolates demonstrated expected growth in both solid and liquid media. In assessing inactivation by the MyT method for the 42 filamentous fungal isolates, no growth was observed over a 28-day incubation period (–/–/– pattern in Table 1) for all isolates, with the exception of 4/42 (9.5%) isolates that showed initial breakthrough: *Purpureocillium lilacinum*, which exhibited breakthrough growth at 2 weeks (–/+/–; later resolved) in liquid broth, *Nigrospora sphaerica*, which exhibited breakthrough growth at 1 week (+/–/–) on solid media, a *Trichophyton* spp. isolate with breakthrough growth at 3 weeks (–/–/+; persistent) in liquid broth, and *Histoplasma capsulatum* with breakthrough growth at 2 weeks (–/+/–) in liquid broth. All four fungal isolates were re-inoculated using the MyT method and were resolved with no growth (final 1/42; 2.4%) up to 4 weeks (best two of three replicates), except for the *Trichophyton* spp., which continued to grow despite multiple re-inoculations.

### MALDI-TOF performance

Fungal identifications by the different methodologies and media types were assessed by MALDI-TOF MS spectral scores. For each isolate, methodology, and media type, specimens were run in triplicate by the same trained operator, and the average of the three results was measured against a pre-specified threshold. A cut-off score of 2.00 was used for species-level identification, 1.70 for genus-level identification, and isolates yielding scores less than 1.70 or failing to generate spectral peaks were classified as no identification possible (Table 2). Overall, the MyT method had the highest rate of successful fungal identifications (90.5% and 92.9% from routine media and IDFP, respectively) compared to both the NIH extraction method (61.9% and 81.0% from routine media and IDFP, respectively) and the CDC extraction method (66.7% and 52.4% from routine media and IDFP, respectively). The MyT method also produced the highest proportion of species-level identification, with 61.9% of isolates identified to the species level from routine fungal media, and 69.1% identified to the species level from IDFP. Similarly, the MyT method also produced the highest average identification scores of 2.03 and 2.06 from routine media and IDFP, respectively (Fig. 1). While general trends of MALDI-TOF MS scores could be visibly observed, only two significant differences were identified when comparing the six different method iterations: the MyT method (IDFP) vs. CDC extraction method (routine media; $P = 0.0421$) and the MyT method (IDFP) vs. CDC extraction method (IDFP; $P = 0.0049$).

Comparison of the different extraction methods was further analyzed by stratifying the molds by their different clinical/mycologic classifications and assessing spectral scores (Table 3). We observed that the MyT method produced the highest average scores (routine media and IDFP) and was also the only extraction method to achieve 100% fungal identifications of specific clinical groups (100% identification of all dermatophytes and mucoraceous molds from routine media and 100% identification of all dematiaceous, dermatophytic, and mucoraceous molds from IDFP), with the exception of the CDC extraction method performed from IDFP, which successfully identified all dermatophytes. The use of Conidia IDFP further enhanced identification accuracy and spectra scores for both the MyT method and NIH extraction; however, a decrease in accuracy (66.7% vs 52.4%) and average MALDI-TOF MS score (1.78 vs 1.77) was observed for the CDC extraction method.

**TABLE 1** Assessment of fungal inactivation using the mycelium transfer (MyT) method across 42 unique fungal specimens, as performed in triplicate in both liquid and solid media[e]

| | Control | Liquid broth | Plated media |
|---|---|---|---|
| **Hyaline molds** | | | |
| *Aspergillus brasiliensis* | (+) | (−/−/−) | (−/−/−) |
| *Aspergillus flavus* | (+) | (−/−/−) | (−/−/−) |
| *Aspergillus fumigatus* | (+) | (−/−/−) | (−/−/−) |
| *Aspergillus niger* | (+) | (−/−/−) | (−/−/−) |
| *Aspergillus terreus* | (+) | (−/−/−) | (−/−/−) |
| *Aspergillus ustus* | (+) | (−/−/−) | (−/−/−) |
| *Aspergillus versicolor* | (+) | (−/−/−) | (−/−/−) |
| *Fusarium equiseti* | (+) | (−/−/−) | (−/−/−) |
| *Fusarium oxysporum* | (+) | (−/−/−) | (−/−/−) |
| *Fusarium solani* | (+) | (−/−/−) | (−/−/−) |
| *Penicillium chrysogenum* | (+) | (− −/−) | (− −/−) |
| *Penicillium citreonigrum* | (+) | (−/−/−) | (− −/−) |
| *Purpureocillium lilacinum* | (+) | (−/+/−)[a] | (−/−/−) |
| *Scedosporium* spp. | (+) | (−/−/−) | (−/−/−) |
| *Scopulariopsis brevicaulis* | (+) | (−/−/−) | (−/−/−) |
| *Scopulariopsis gracilis* | (+) | (−/−/−) | (−/−/−) |
| *Talaromyces rugulosus* | (+) | (−/−/−) | (−/−/−) |
| **Dematiaceous molds** | | | |
| *Alternaria alternata* | (+) | (−/−/−) | (−/−/−) |
| *Alternaria pullulans* | (+) | (−/−/−) | (−/−/−) |
| *Chaetomium globosum* | (+) | (−/−/−) | (−/−/−) |
| *Cladophilaphora* spp. | (+) | (−/−/−) | (−/−/−) |
| *Cladosporium cladosporioides* | (+) | (−/−/−) | (−/−/−) |
| *Curvularia geniculata* | (+) | (−/−/−) | (−/−/−) |
| *Curvularia* spp. | (+) | (−/−/−) | (−/−/−) |
| *Epicoccum nigrum* | (+) | (−/−/−) | (−/−/−) |
| *Epicoccum* spp. | (+) | (−/−/−) | (−/−/−) |
| *Nigrospora sphaerica* | (+) | (−/−/−) | (+/−/−)[b] |
| *Phoma herbarum* | (+) | (−/−/−) | (−/−/−) |
| *Ulocladium* spp. | (+) | (−/−/−) | (−/−/−) |
| **Dermatophytes** | | | |
| *Trichophyton interdigitale* | (+) | (−/−/−) | (−/−/−) |
| *Trichophyton mentagrophytes* | (+) | (−/−/−) | (−/−/−) |
| *Trichophyton rubrum* | (+) | (−/−/−) | (−/−/−) |
| *Trichophyton tonsurans* | (+) | (−/−/−) | (−/−/−) |
| *Trichophyton* spp. | (+) | (/−/−+)[c] | (−/−/−) |
| **Mucoraceous molds** | | | |
| *Lichtheimia ramosa* | (+) | (−/−/−) | (−/−/−) |
| *Mucor circinelloides* | (+) | (−/−/−) | (−/−/−) |
| *Rhizopus oryzae* | (+) | (−/−/−) | (−/−/−) |
| **Dimorphic molds** | | | |
| *Blastomyces dermatitidis* | (+) | (−/−/−) | (−/−/−) |
| *Blastomyces gilchristii* | (+) | (−/−/−) | (−/−/−) |
| *Coccidioides immitis* | (+) | (−/−/−) | (−/−/−) |
| *Coccidioides posadasii* | (+) | (−/−/−) | (−/−/−) |

TABLE 1 Assessment of fungal inactivation using the mycelium transfer (MyT) method across 42 unique fungal specimens, as performed in triplicate in both liquid and solid media[e] (Continued)

|  | Control | Liquid broth | Plated media |
|---|---|---|---|
| *Histoplasma capsulatum* | (+) | (−/+/−)[d] | (−/−/−) |

[a]Breakthrough growth observed 1/3 replicates at 2 weeks incubation, and repeat testing yielded 0/3 and 0/3 breakthrough growth through 4 weeks.
[b]Breakthrough growth observed 1/3 replicates at 1 week of incubation, and repeat testing yielded 0/3 and 0/3 breakthrough growth through 4 weeks.
[c]Breakthrough growth observed 2/3 replicates at 3 weeks of incubation, and breakthrough growth was still observed with repeat testing, yielding 1/3 and 2/3 breakthrough growth at 3 weeks of incubation.
[d]Breakthrough growth observed 1/3 replicates at 2 weeks incubation, and repeat testing yielded 0/3 and 0/3 breakthrough growth through 4 weeks.
[e]spp., species; +, growth; −, no growth.

Three fungal isolates failed to be consistently identified across the six different method iterations. The first, *Fusarium equiseti*, is a hyaline mold that was initially isolated from a patient wound culture and identified morphologically by standard-of-care (SOC) workup and then confirmed through send-out testing by sequencing of the internal transcribed spacer (ITS)-1 and ITS-2 genes. The other two isolates were dimorphic molds, *Coccidioides immitis* and *Coccidioides posadasii*, both of which were isolated from patients with travel history to endemic regions. Both isolates were originally presumptively identified based on fungal morphology through SOC workup and diagnostically confirmed with quantitative enzyme immunoassay from patient serum. Species differentiation for the *Coccidioides* isolates was determined by real-time PCR using a previously published assay (13). Of note, while *F. equiseti* is described in the Bruker MBT Filamentous Fungi database, the inability of these isolates to be identified by MALDI-TOF MS was attributed to the absence of *Coccidioides* in the database utilized in this study.

## DISCUSSION

Adequate fungal inactivation is crucial to both successful identification by MALDI-TOF MS and ensuring the safety of laboratory personnel. Standard cell disruption methods are often limited by the presence of more robust cell walls in fungi, as compared to bacteria, necessitating the use of extensive cell lysis approaches for improved protein extraction (14), often requiring multiple steps of washing, chemical inactivation, and protein extraction (10, 15). Full extraction methods are effective but time intensive, requiring several centrifugation and wash steps. The NIH rapid extraction method uses zirconia-silica beads and an extraction solution, omitting washing and ethanol inactivation steps to achieve a 5-minute processing time and is a substantial improvement from previous procedures, which took anywhere from 35 minutes to over an hour (10, 16, 17). Alternatively, the MyT method, which utilizes a formic acid sandwich approach, has demonstrated comparable efficacy with improved efficiency (15). Among the methods evaluated in this study, the MyT method emerged as the most accurate and sensitive in the identification of filamentous fungi via MALDI-TOF MS; this method yielded the highest average spectral scores, the greatest proportion of species-level identifications, and significantly reduced the amount of hands-on time required by laboratory personnel. Proper inactivation of fungal isolates is critically important to laboratory safety, particularly in the case of dimorphic molds. While only five unique

TABLE 2 Fungal identification accuracy by matrix-assisted laser desorption/ionization–time of flight mass spectrometry (MALDI-TOF MS) using different extraction methods across routine and specialized media[a,b]

| Identification | Routine media | | | ID fungal plates | | |
|---|---|---|---|---|---|---|
|  | MyT | NIH extraction | CDC extraction | MyT | NIH extraction | CDC extraction |
|  | *n* (%) | *n* (%) | *n* (%) | *n* (%) | *n* (%) | *n* (%) |
| Species level | 26/42 (61.9%) | 16/42 (38.1%) | 10/42 (23.8%) | 29/42 (69.1%) | 19/42 (45.2%) | 14/42 (33.3%) |
| Genus level | 12/42 (28.6%) | 10/42 (23.8%) | 18/42 (42.9%) | 10/42 (23.8%) | 15/42 (35.7%) | 8/42 (19.0%) |
| No identification | 4/12 (9.5%) | 16/42 (38.1%) | 14/42 (33.3%) | 3/42 (7.1%) | 8/42 (19.0%) | 20/42 (47.6%) |

[a]The cut-off scores for species- and genus-level identification were ≥2.00 and 1.7–1.99, respectively. Obtained scores <1.70 were considered as "no identification possible".
[b]MyT, mycelium transfer; NIH, National Institutes of Health; CDC, Centers for Disease Control and Infection Prevention.

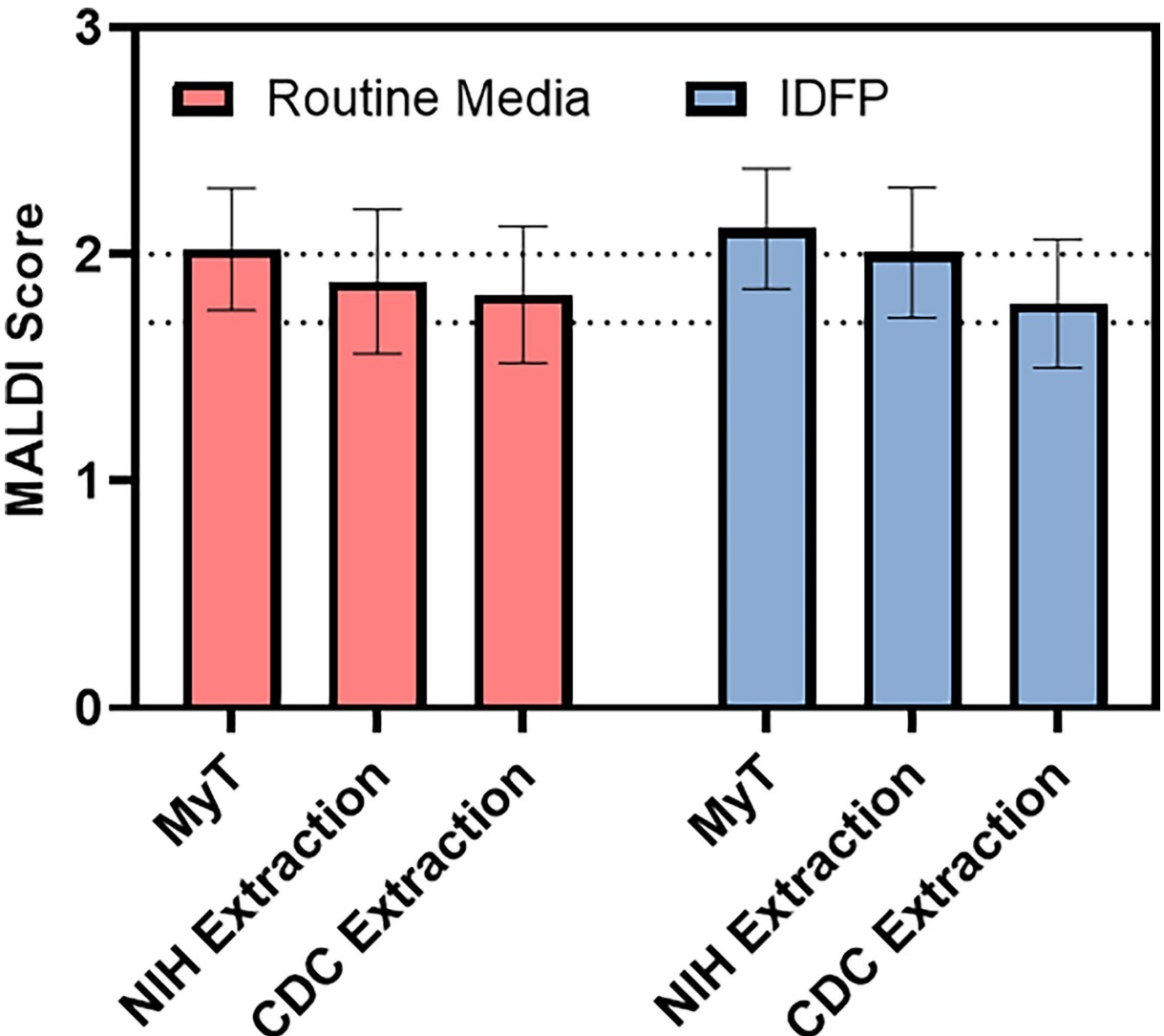

**FIG 1** Analysis of extraction efficiency and matrix-assisted laser desorption/ionization-time of flight mass spectrometry (MALDI-TOF MS) scores across different media. Box: average score; whiskers: standard deviation; intersecting horizontal lines at MALDI-TOF MS score = 2.00 (species-level identification) and 1.70 (genus-level identification); IDFP, Conidia ID fungal plates; MyT, mycelium transfer; NIH, National Institutes of Health; CDC, Centers for Disease Control and Prevention.

dimorphic fungal species were assessed in this study, a total of 40 unique dimorphic fungal specimens were assessed as part of the laboratory validation (data not shown). Importantly, we observed breakthrough growth with one of the *Trichophyton* spp. included in the inactivation arm of the study. While the MyT method was overall effective in inactivating most isolates, given the potential for breakthrough growth, biosafety precautions should always be maintained in accordance with required containment barriers used for setting up MALDI-TOF MS (i.e., PPE, BSC, and appropriate BSL-2 or BSL-3 practices) (8). Overall, our data suggest that the MyT method, while lacking a full extraction procedure, sufficiently inactivates most fungal isolates, reducing safety concerns for laboratory personnel.

Commercially available MALDI-TOF MS platforms include systems from Hardy Diagnostics, Bruker, and bioMérieux, with only the latter two possessing Food and Drug Administration (FDA) clearance for their instruments. Accurate fungal diagnosis by MALDI-TOF MS is dependent on adequate databases for spectral data comparison. Of note, the bioMérieux Vitek MS V3.2 is the only FDA-cleared database for use that

**TABLE 3** Mass-spectrometry laser desorption/ionization-time of flight mass spectrometry (MALDI-TOF MS) genus identification outcomes by extraction method across key clinical fungal groups[a]

| Identification[b] | MyT<br>n (%) | Average Score | NIH extraction<br>n (%) | Average score | CDC extraction<br>n (%) | Average score |
|---|---|---|---|---|---|---|
| Routine media | | | | | | |
| Hyaline | 16/17 (94.1) | 2.21 | 13/17 (76.5) | 2.03 | 14/17 (82.4) | 1.86 |
| Dematiaceous | 11/12 (91.7) | 1.96 | 6/12 (50.0) | 1.52 | 6/12 (50.0) | 1.78 |
| Dermatophytes | 5/5 (100.0) | 2.10 | 3/5 (60.0) | 1.91 | 2/5 (40.0) | 1.74 |
| Mucorales | 3/3 (100.0) | 1.82 | 2/3 (66.7) | 1.83 | 1/3 (33.3) | 1.78 |
| Dimorphic | 3/5 (60.0) | 2.06 | 2/5 (40.0) | 1.72 | 2/5 (40.0) | 1.76 |
| Overall | 38/42 (90.5) | 2.03 | 26/42 (61.9) | 1.80 | 28/42 (66.7) | 1.78 |
| ID fungal plates | | | | | | |
| Hyaline | 16/17 (94.1) | 2.15 | 16/17 (94.1) | 2.04 | 6/17 (35.3) | 1.73 |
| Dematiaceous | 12/12 (100.0) | 2.16 | 9/12 (75.0) | 1.96 | 8/12 (66.7) | 1.80 |
| Dermatophytes | 5/5 (100.0) | 2.2 | 4/5 (80.0) | 2.09 | 5/5 (100.0) | 1.99 |
| Mucorales | 3/3 (100.0) | 1.92 | 2/3 (66.7) | 1.76 | 1/3 (33.3) | 1.65 |
| Dimorphic | 3/5 (60.0) | 1.88 | 3/5 (60.0) | 1.85 | 2/5 (40.0) | 1.70 |
| Overall | 39/42 (92.9) | 2.06 | 34/42 (81.0) | 1.94 | 22/42 (52.4) | 1.77 |

[a]MyT, mycelium transfer; NIH, National Institutes of Health; CDC, Centers for Disease Control and Infection Prevention.
[b]Identification determined minimum genus-level identification.

includes a library for filamentous fungi, while the Bruker MBT Filamentous Fungi Library is commercially available as RUO. The NIH also has a filamentous mold database, encompassing 76 genera and 152 species, which includes 10 dimorphic fungi (3). The inclusion of dimorphic fungi in the reference library is an important consideration given regulatory requirements that certain clinical laboratories must uphold. In accordance with the College of American Pathologists guidelines, all suspected dimorphic fungal isolates require confirmatory identification using an additional methodology beyond traditional microscopy; this can include exo-antigen, molecular, yeast-mold conversion, tissue phase detection, or MALDI-TOF MS (8).

In our assessment, the MyT method generally outperformed both rapid extraction methods. One potential explanation for this is that multiple intervention steps in fungal extractions result in lower quality proteins for spectral analysis (18). Another possibility is a potential dilution effect with extractions, compared to the MyT method, which transfers all the fungal mass directly to the plate target. The incorporation of Conidia IDFP generally improved identification scores for all extraction methods compared to routine media. The plate includes a transparent membrane on the surface of the agar, which allows for the growth of molds but restricts fungal hyphae, facilitating a clean mycelium harvest for MALDI-TOF MS. While effective in improving identification scores, laboratories should consider a cost-benefit analysis before incorporating these plates as part of their routine mycology workflows. Manufacturer availability, whether these plates should be added or replace routine media, and the significance of score improvement should all be assessed prior to implementation. In our study, we found that the implementation of the IDFP with the MyT method insignificantly improved our average identification score from 2.03 to 2.06 ($P > 0.9999$), both of which are above the cut-off threshold for species-level identification; in this case, the added financial cost of incorporating IDFP may outweigh the benefit of score improvement.

Our study is not without limitations; one limitation includes that this was a single-site study where our performance findings may not be generalizable. Several variables, such as colony age, fungal mass, and technologist skill and expertise, may contribute to variable results and identification scores. Colony age was organism dependent, averaging a couple of weeks for optimal growth. While technologists were trained, biomass size was also likely variable, but this limitation is not unique to fungi. Furthermore, operator-stratified performance data were not captured prospectively but would be interesting to evaluate in future studies. Another limitation of this study was the sample size; in selecting specimens, we chose only to include unique fungal

species ($n$ = 42), as opposed to unique clinical specimens. The rationale for this was to prioritize assessing the inactivation of the MyT method and taking cost considerations into account with IDFP and methodology comparisons. Formic acid concentration and biomass size were not systematically manipulated during kill studies but would be of interest for future studies. The restriction of only using the Bruker platform and software, due to instrument availability at our site, was another limitation of this study. Identification scores and performance may be very different when assessed on a different platform or with a different database (19–21). Furthermore, the restriction of isolate availability in the Bruker MBT Filamentous Fungi library was apparent to us (i.e., *Coccidioides* species spectral profiles unavailable) and could further be a limitation to laboratories that employ this methodology.

Overall, the MyT method is a rapid pre-processing, extraction-free protocol for the identification of filamentous fungi using MALDI-TOF MS with improved spectral scores. Performance may also be enhanced with the use of IDFP but is not strictly necessary for successful fungal identification. We believe that the implementation of this methodology will allow clinical laboratories to safely and effectively incorporate filamentous fungi identification into their routine workflows without the labor-intensiveness of both rapid and full fungal extraction procedures.

## AUTHOR AFFILIATIONS

[1]Department of Pathology and Laboratory Medicine, Indiana University School of Medicine, Indianapolis, Indiana, USA

[2]Division of Clinical Microbiology and Serology, Eskenazi Health, Indianapolis, Indiana, USA

## AUTHOR ORCIDs

Kenneth Gavina ⓘ http://orcid.org/0000-0001-8358-5611

## AUTHOR CONTRIBUTIONS

Halimat Olaniyan, Formal analysis, Writing – original draft, Writing – review and editing | Chris Kendra, Supervision, Writing – review and editing | Alexandria Dugan, Formal analysis, Methodology, Validation | Shelby Kruer, Methodology, Validation | Alisha Zuber, Methodology, Validation | Jessica Bywaters, Methodology, Supervision, Validation | Amber Ryan, Methodology, Project administration, Supervision, Writing – review and editing | Kenneth Gavina, Conceptualization, Formal analysis, Investigation, Methodology, Resources, Supervision, Validation, Writing – original draft, Writing – review and editing

## ETHICS APPROVAL

Ethical approval was obtained from the Institutional Review Board (IRB#26405) at the Indiana University School of Medicine.

## ADDITIONAL FILES

The following material is available online.

Open Peer Review

**PEER REVIEW HISTORY (review-history.pdf).** An accounting of the reviewer comments and feedback.

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
