## [Reviewer comments · Microbiology Spectrum]

Microbiology Spectrum

Evaluation of multiple extraction and inactivation methods for the rapid identification of filamentous fungi using matrix-assisted laser desorption/ionization time-of-flight mass spectrometry

Halimat Olaniyan, Christopher Kendra, Alexandria Dugan, Shelby Kruer, Alisha Zuber, Jessica Bywaters, Amber Ryan, and Kenneth Gavina

Corresponding Author(s): Kenneth Gavina, Indiana University School of Medicine

Review Timeline:

Submission Date:	December 2, 2025
Editorial Decision:	December 29, 2025
Revision Received:	February 16, 2026
Accepted:	February 17, 2026

Editor: Paul Luethy

Reviewer(s): Disclosure of reviewer identity is with reference to reviewer comments included in decision letter(s). The following individuals involved in review of your submission have agreed to reveal their identity: Anna Frances Lau (Reviewer #1)

Transaction Report:

DOI: <https://doi.org/10.1128/spectrum.03859-25>

Re: Spectrum03859-25 (Evaluation of multiple extraction and inactivation methods for the rapid identification of filamentous fungi using matrix-assisted laser desorption/ionization time-of-flight mass spectrometry)

Dear Dr. Kenneth Gavina:

Thank you for the privilege of reviewing your work. Below you will find my comments, instructions from the Spectrum editorial office, and the reviewer comments.

Revision Guidelines

Sincerely,
Paul Luethy
Editor
Microbiology Spectrum

Reviewer #1 (Comments for the Author):

The manuscript by Olaniyan et al describes an evaluation of 42 unique mold species comparing the performance of 3 different extraction methods (MyT, NIH rapid, and CDC) for identification accuracy using the Bruker RUO FilFungal database. The authors go the extra length to evaluate inactivation effectiveness of the MyT method, which has not been widely assessed previously and has caused some hesitation in the field for adopting this method due to biosafety and cross-contamination

concerns. The manuscript is well-written and fills a gap in the field. Some comments below:

Major comments:

1. Lines 109-117 - please check references carefully. Reference 3 is a full extraction method from the NIH. Reference 10 is a rapid extraction method from the NIH. Please provide reference for the MyT method and ensure it is clear that the MyT method is a manufacturer recommended method.
2. Lines 131-133 and Table 1 - how many isolates had breakthrough growth? Line 133 states that the best out of 3 was recorded so one assumes that Table 1 only shows the best out of 3 results. Raw data would be helpful (-/- or +/- etc). This is confusing with Lines 162-165 which indicates that only 4 had breakthrough growth, 3 of which resolved after repeat testing. Please provide overall data on inactivation success that includes repeat testing (xx/42+repeats had breakthrough; inverse is overall success rate. Can this be used to make a risk-based decision for application of MyT in the clinical setting - see comments 7 and 8 below).
3. Table 3: Please clarify whether data included in Table 3 was for successful Genus level identification and better? What are results if Table 3 only assesses successful species level identification?
4. Can the authors confirm that all identifications were accurate? ie no misidentifications at the species or genus level?
5. Line 195: was *Fusarium equiseti* in the database?
6. Lines 232-233 - can the authors analyze the spectra against a different database (NIH or MSI) to see if the *Coccidioides* isolates can be identified?
7. Line 223: Can the authors discuss containment barriers further? The mass spec instrument is typically in an open lab. What if the plate drops? Staff do not typically wear masks in an open BSL2 lab. Can the authors elaborate on occupational hazards, immunocompromised staff etc?
8. As discussed in lines 81-83, mold inactivation (particularly of dimorphs) is important. Only 3 dimorphic molds were assessed in this study. How confident are the authors that dimorphs can be adequately inactivated via the MyT method? Perhaps language can be softened and/or this can be added to the limitations section.
9. Line 135: Please provide more details on MALDI. Single, duplicate, or triplicate spots? If multiple, how was identification determined?
10. Lines 132-133 and 224-225: would adjusting the formic acid concentration or volume assist in preventing breakthroughs?
11. Lines 255-256: The authors highlight several variables in this study that suggest that successful inactivation via the MyT method may be operator dependent - this raises concerns for biosafety. It also suggests that extraction methods may not have been fairly compared. Suggest providing more data to alleviate concerns. What was the variable in colony age? Were all 3 extraction methods performed on the same plate/same age of colony? Please elaborate on fungal mass and which extraction method more/less mass would have affected. Please evaluate results based on technologists. Was there mostly one person conducting this study?

Reviewer #2 (Comments for the Author):

Review of Evaluation of multiple extraction and inactivation methods for the rapid identification of 2 filamentous fungi using matrix-assisted laser desorption/ionization time-of-flight mass 3 spectrometry

The manuscript addresses an important and clinically relevant question regarding the safety and performance of MALDI-TOF MS based identification of filamentous fungi. The study is well designed in terms of diverse set of clinically relevant molds as well as evaluating multiple extraction methods along with inactivation. Though, I think some conclusions may be a bit overstated.

Major Changes:

- Using best two of three replicates for inactivation studies introduces bias and likely overestimates safety. From a biosafety standpoint, any breakthrough growth should be considered a point-of-failure where extra precautions are needed. Excluding such data points isn't ideal. 4/42 isolates initially showed breakthrough growth which is higher than the extraction methods tested. Especially for the one *Trichophyton* that persistently failed, which clearly shows that MyT does not guarantee complete inactivation compared to tube extraction methods.
- Lines 221-224 *Trichophyton* species should not be described as "low-risk" ---- the risk should not be minimized rather it should warrant caution and/or protocol modification.
- Majority of the studies conclusions are not statistically significant and it should be acknowledged that the differences are descriptive rather than statistically significant.
- Database limitations are a significant issue when comparing the different methods - I wonder if study would be stronger in excluding *Coccidioides* from performance metrics. If the organism is not identifiable in the database, then it can't truly evaluate the protocol. However, it may be an anecdotal to include that even when something not present in database it doesn't result in a misidentification.
- Line 118 - ensuring to collect an equivocal sample is confusing and maybe subjective. Is the evaluation of the protocols, based on the same biomass because the different methods understandably need varying biomass which can be stated as a limitation. If the study was designed to use the same biomass for all extractions, then it would be helpful to include a measurable standard ---- 1cm of colony area etc.

Minor changes:

- Line 109 - T missing from MALDI-TOF

- Line 115 - Rapid MyT is that different from MyT
- Line 235 - Citing the CAP reference may be helpful here as there may be multiple interpretations
- Line 118 - Is equivocal correct or is that meant to be equivalent
- Line 225 - Given the lack of consistent inactivation, this statement is overstated.

Review of Evaluation of multiple extraction and inactivation methods for the rapid identification of 2 filamentous fungi using matrix-assisted laser desorption/ionization time-of-flight mass 3 spectrometry

The manuscript addresses an important and clinically relevant question regarding the safety and performance of MALDI-TOF MS based identification of filamentous fungi. The study is well designed in terms of diverse set of clinically relevant molds as well as evaluating multiple extraction methods along with inactivation. Though, I think some conclusions may be a bit overstated.

Major Changes:

- Using best two of three replicates for inactivation studies introduces bias and likely overestimates safety. From a biosafety standpoint, any breakthrough growth should be considered a point-of-failure where extra precautions are needed. Excluding such data points isn't ideal. 4/42 isolates initially showed breakthrough growth which is higher than the extraction methods tested. Especially for the one Trichophyton that persistently failed, which clearly shows that MyT does not guarantee complete inactivation compared to tube extraction methods.
- Lines 221-224 Trichophyton species should not be described as "low-risk" ---- the risk should not be minimized rather it should warrant caution and/or protocol modification.
- Majority of the studies conclusions are not statistically significant and it should be acknowledged that the differences are descriptive rather than statistically significant.
- Database limitations are a significant issue when comparing the different methods – I wonder if study would be stronger in excluding Coccidioides from performance metrics. If the organism is not identifiable in the database, then it can't truly evaluate the protocol. However, it may be an anecdotal to include that even when something not present in database it doesn't result in a misidentification.
- Line 118 – ensuring to collect an equivocal sample is confusing and maybe subjective. Is the evaluation of the protocols, based on the same biomass because the different methods understandably need varying biomass which can be stated as a limitation. If the study was designed to use the same biomass for all extractions, then it would be helpful to include a measurable standard ---- 1cm of colony area etc.

Minor changes:

- Line 109 - T missing from MALDI-TOF
- Line 115 – Rapid MyT is that different from MyT
- Line 235 – Citing the CAP reference may be helpful here as there may be multiple interpretations
- Line 118 – Is equivocal correct or is that meant to be equivalent
- Line 225 – Given the lack of consistent inactivation, this statement is overstated.

Response: We thank the reviewers for their constructive feedback. Below, we provide responses and describe appropriate changes implemented in the revised manuscript. Line numbers refer to the updated version entitled, not version with tracked changes.

Reviewer #1 (Comments for the Author):

The manuscript by Olaniyan et al describes an evaluation of 42 unique mold species comparing the performance of 3 different extraction methods (MyT, NIH rapid, and CDC) for identification accuracy using the Bruker RUO FilFungal database. The authors go the extra length to evaluate inactivation effectiveness of the MyT method, which has not been widely assessed previously and has caused some hesitation in the field for adopting this method due to biosafety and cross-contamination concerns. The manuscript is well-written and fills a gap in the field. Some comments below:

Major comments:

1. Lines 109-117 - please check references carefully. Reference 3 is a full extraction method from the NIH. Reference 10 is a rapid extraction method from the NIH. Please provide reference for the MyT method and ensure it is clear that the MyT method is a manufacturer recommended method.

Response: Thank you for your feedback. The Mycelium Transfer (MyT) workflow is a Bruker-supported workflow inspired by extended direct transfer and uses formic acid pretreatment of the applicator before mycelium transfer. The references on lines 111-114 have been updated to reflect this.

2. Lines 131-133 and Table 1 - how many isolates had breakthrough growth? Line 133 states that the best out of 3 was recorded so one assumes that Table 1 only shows the best out of 3 results. Raw data would be helpful (-/-/- or +/-/- etc). This is confusing with Lines 162-165 which indicates that only 4 had breakthrough growth, 3 of which resolved after repeat testing. Please provide overall data on inactivation success that includes repeat testing (xx/42+repeats had breakthrough; inverse is overall success rate. Can this be used to make a risk-based decision for application of MyT in the clinical setting - see comments 7 and 8 below).

Response: We agree with the reviewer's comments and revised Table 1 to include initial and final inactivation outcomes with raw replicate patterns for all the isolates including the 4/42 isolates (9.5%) with initial breakthrough growth. After repeat inoculation using the MyT method with two independent operators, three of these resolved (final 1/42; 2.4%). In lines 162-171, we discuss the point of failure rate with any breakthrough and the final resolved rate. While MyT was generally successful in inactivating specimens, appropriate safety precautions should be maintained regardless in case of breakthrough growth. We further clarified safety precautions

taken by lab personnel running these tests (separate area of the lab in a biosafety cabinet) in lines 121-128.

3. Table 3: Please clarify whether data included in Table 3 was for successful Genus level identification and better? What are results if Table 3 only assesses successful species level identification?

Response: We have modified the title for Table 3 to more clearly indicate successful identification at genus-level. Species versus genus level identification was highlighted in Table 2 for the different extraction levels.

4. Can the authors confirm that all identifications were accurate? ie no misidentifications at the species or genus level?

Response: Thank you for the comment - yes, all identifications were accurate except for the three organisms described from lines 202-212: *Fusarium equiseti*, *Coccidioides immitis* and *C. Posadasii*.

5. Line 195: was *Fusarium equiseti* in the database?

Response: Per Bruker manufacturer *Fusarium equiseti* is included in their library. We've modified line 210 to clarify.

6. Lines 232-233 - can the authors analyze the spectra against a different database (NIH or MSI) to see if the *Coccidioides* isolates can be identified?

Response: Cross-evaluation against NIH or MSI databases were outside the scope of this study given Bruker was the only available platform at our institution. We acknowledge this limitation in the discussion and have updated the references to cite literature demonstrating the strengths and limitations of various databases in lines 278-283.

7. Line 223: Can the authors discuss containment barriers further? The mass spec instrument is typically in an open lab. What if the plate drops? Staff do not typically wear masks in an open BSL2 lab. Can the authors elaborate on occupational hazards, immunocompromised staff etc?

Response: Thank you for this comment, we've expanded on the methods section to describe biosafety controls in place in lines 121-123. All fungal related testing including the MALDI-TOF instrument is confined to a secure area of the lab (BSL2, sealed plates, safety cabinets for all manipulation) with limited personnel assigned to rotate through. All specimens are also

manipulated and prepped for analysis under a safety hood to further limit potential exposure. Considerations for immunocompromised personnel were consistent with institutional occupational health policies. This clarification has been added to the paper in lines 127-128.

8. As discussed in lines 81-83, mold inactivation (particularly of dimorphs) is important. Only 3 dimorphic molds were assessed in this study. How confident are the authors that dimorphs can be adequately inactivated via the MyT method? Perhaps language can be softened and/or this can be added to the limitations section.

Response: We thank the reviewer for this comment. While only three genera (5 unique spp.) of dimorphs were assessed, they represent the 5 species of dimorphs that account for the majority of dimorphic molds encountered in US laboratories. Additionally, as part of the validation for implementation in the laboratory, a total of 40 unique dimorphic specimens were assessed for fungal inactivation. We've added Lines 229-231 to include this information and softened the language in the discussion.

9. Line 135: Please provide more details on MALDI. Single, duplicate, or triplicate spots? If multiple, how was identification determined?

Response: We added details to lines 128 & 175 to clarify that samples were run in triplicates per operator with both solid and liquid media. The average score across the triplicates was used to decide if the result met the pre-specified threshold (cutoff score of 2 for species level and 1.7 for genus level) with concordant taxonomy across triplicate spots. This was to minimize the impact of imprecision. Discordant triplicate results were resolved by repeating the run. This has been clarified in lines 136-138.

10. Lines 132-133 and 224-225: would adjusting the formic acid concentration or volume assist in preventing breakthroughs?

Response: Thank you for this comment, we used a formic acid concentration at 70% based on established/published methods and did not systematically vary concentration or volume. This has been addressed as a limitation of our study and updated in lines 276-277.

11. Lines 255-256: The authors highlight several variables in this study that suggest that successful inactivation via the MyT method may be operator dependent - this raises concerns for biosafety. It also suggests that extraction methods may not have been fairly compared. Suggest providing more data to alleviate concerns. What was the variable in colony age? Were

all 3 extraction methods performed on the same plate/same age of colony? Please elaborate on fungal mass and which extraction method more/less mass would have affected. Please evaluate results based on technologists. Was there mostly one person conducting this study?

Response: We thank the reviewer for these suggestions. Colony age varied by organism as some isolates took longer to grow sufficient hyphae but was on average a couple of weeks. Technologists running the tests were all trained on how to optimize biomass collection with standardized harvest from the colony edge, targeting approximately 1 cm on solid media or equivalent mycelial mass from liquid broth. For each isolate, all three extraction methods were performed on contemporaneous cultures from the same plate on the same run by the same operator. Operator-stratified performance data were not captured prospectively, and target biomass was not systematically manipulated for evaluation. We updated this information and added these limitation in line 272.

Reviewer #2 (Comments for the Author):

Review of Evaluation of multiple extraction and inactivation methods for the rapid identification of 2 filamentous fungi using matrix-assisted laser desorption/ionization time-of-flight mass 3 spectrometry

The manuscript addresses an important and clinically relevant question regarding the safety and performance of MALDI-TOF MS based identification of filamentous fungi. The study is well designed in terms of diverse set of clinically relevant molds as well as evaluating multiple extraction methods along with inactivation. Though, I think some conclusions may be a bit overstated.

Major Changes:

- Using best two of three replicates for inactivation studies introduces bias and likely overestimates safety. From a biosafety standpoint, any breakthrough growth should be considered a point-of-failure where extra precautions are needed. Excluding such data points isn't ideal. 4/42 isolates initially showed breakthrough growth which is higher than the extraction methods tested. Especially for the one Trichophyton that persistently failed, which clearly shows that MyT does not guarantee complete inactivation compared to tube extraction methods.

Response: We agree with the reviewer and have reframed the biosafety analysis to report the initial point of failure rate (4/42; 9.5%) and the final resolved rate after repeats (41/42; 97.6%) in lines 164-171. We also updated Table 1 with raw replicate patterns.

- Lines 221-224 Trichophyton species should not be described as "low-risk" ---- the risk should not be minimized rather it should warrant caution and/or protocol modification.

Response: Thank you for this feedback. We included a risk-based statement emphasizing safety protocols should be utilized regardless of organism and inactivation method in case of breakthrough growth lines 232-236.

- Majority of the studies conclusions are not statistically significant and it should be acknowledged that the differences are descriptive rather than statistically significant.

Response: We updated the statistical analysis and results to emphasize descriptive trends and explicitly state where comparisons reached statistical significant.

- Database limitations are a significant issue when comparing the different methods - I wonder if study would be stronger in excluding *Coccidioides* from performance metrics. If the organism is not identifiable in the database, then it can't truly evaluate the protocol. However, it may be an anecdotal to include that even when something not present in database it doesn't result in a misidentification.

Response: Thank you for these comments, we included results from *Coccidioides* isolates as there were two arms to this study (1) addressing the effectiveness of Myt for inactivation which it did successfully inactive *Coccidioides* isolates and (2) assessing the performance of isolates using MALDI-TOF which our Bruker library was unable to assess. We have emphasized the study database limitations in regards to *Coccidioides* in lines 280-282.

- Line 118 - ensuring to collect an equivocal sample is confusing and maybe subjective. Is the evaluation of the protocols, based on the same biomass because the different methods understandably need varying biomass which can be stated as a limitation. If the study was designed to use the same biomass for all extractions, then it would be helpful to include a measurable standard ---- 1cm of colony area etc.

Response: We agree with the reviewer; technologists running the tests were all trained on how to optimize biomass collection with standardized harvest from the colony edge, targeting approximately 1 cm on solid media or equivalent mycelial mass from liquid broth. Biomass was not varied and this clarification as well as limitation have been updated in lines 99-101 & 129-130.

Minor changes:

- Line 109 - T missing from MALDI-TOF

Response: Thank you for catching this, it has been updated on line 111.

- Line 115 - Rapid MyT is that different from MyT

Response: Rapid Myt and MyT are the same. We have made edits to only refer to this method as MyT consistently throughout the paper.

- Line 235 - Citing the CAP reference may be helpful here as there may be multiple interpretations

Response: The CAP reference has been cited in line 236 for added clarity.

- Line 118 - Is equivocal correct or is that meant to be equivalent

Response: Thank you for pointing this out. We meant equivalent and have updated line 101 accordingly.

- Line 225 - Given the lack of consistent inactivation, this statement is overstated.

Response: We agree and have softened the language on line 233-237.

Re: Spectrum03859-25R1 (Evaluation of multiple extraction and inactivation methods for the rapid identification of filamentous fungi using matrix-assisted laser desorption/ionization time-of-flight mass spectrometry)

Dear Dr. Kenneth Gavina:

Your manuscript has been accepted, and I am forwarding it to the ASM production staff for publication. Your paper will first be checked to make sure all elements meet the technical requirements. ASM staff will contact you if anything needs to be revised before copyediting and production can begin. Otherwise, you will be notified when your proofs are ready to be viewed.

Sincerely,
Paul Luethy
Editor
Microbiology Spectrum